# The Microbiome and Genitourinary Cancers: A New Frontier

**DOI:** 10.3390/cancers17223606

**Published:** 2025-11-08

**Authors:** Timothy B. Winslow, Sophia Gupta, Vedha Sai Vaddaraju, Brendan J. Guercio, Deepak M. Sahasrabudhe

**Affiliations:** Wilmot Cancer Institute, University of Rochester, Rochester, NY 14642, USA; timothy_winslow@urmc.rochester.edu (T.B.W.); sgupta52@u.rochester.edu (S.G.); vedha_vaddaraju@urmc.rochester.edu (V.S.V.);

**Keywords:** bladder cancer, kidney cancer, gut microbiome, immunotherapy resistance, urinary microbiome, short chain fatty acids

## Abstract

**Simple Summary:**

This review article addresses the current understanding of how the bacteria living in the gut and urinary tract interact with our immune system. This has implications not only for development of bladder and kidney cancer as well as how they respond to treatment. Understanding this interplay of the immune system with these bacteria has led to small clinical trials to modulate these interaction for clinical benefit. There additional studies are needed so that the outcomes of cancer patients can be improved.

**Abstract:**

Introduction of immune checkpoint inhibitors and targeted agents has markedly improved outcomes and extended survival in urothelial and renal cell carcinoma. However, the substantial subset of cases are treatment-resistant. Emerging strategies aim to enhance the effectiveness of immunotherapy. One area of growing interest and promise is the microbiome. The microbiome plays a complex and dynamic role in regulating the immune system, and represents a new frontier as a promising target for modulating response to immunotherapy. This review summarizes recent advances in our understanding of the microbiome, its interactions with immunotherapy, novel avenues for microbiome modification, and potential implications for the treatment of urothelial and renal cell carcinoma.

## 1. Introduction

Genitourinary (GU) malignancies are common, with urothelial carcinoma (UC) and renal cell carcinoma (RCC) being the fourth and sixth most common malignancies in men, respectively, and RCC being the nineth most common in women in the United States [1]. UC and RCC have been historically difficult to treat with conventional therapies such as cytotoxic chemotherapy [2,3]. Introduction of immune checkpoint inhibitors (ICIs)—monoclonal antibodies that bind and block the immune checkpoint pathway proteins PD-1, PDL-1 or CTLA-4—has resulted in significant improvements in the survival of patients with UC and RCC, either alone or in combination with other agents such as antibody-drug conjugates or VEGF tyrosine kinase inhibitors [4,5,6,7,8,9,10,11]. Despite these advances, primary and acquired resistance to ICIs remain substantial clinical challenges, resulting in five-year survival for advanced UC and RCC of less than 50% [10,12,13,14,15], underscoring the urgent need to identify the modifiable host and tumor factors that influence therapeutic efficacy [16,17].

Among these, the gut microbiome has emerged as a critical extrinsic regulator of systemic immunity and cancer therapy outcomes. The microbiota consists of trillions of microorganisms—bacteria, viruses, fungi—that reside primarily in the gastrointestinal tract, interacting intricately with the host immune system [18,19]. Understanding gut–immune cross-talk is an area of research that is rapidly expanding. The gut microbiome has not only been implicated in response to therapy in a variety of cancers, including GU malignancies, but also as a modifiable risk factor, and it has also been shown to play a role in carcinogenesis [20,21,22].

The gut microbiome influences response to cancer therapy by modulating systemic immunity. The mechanism of the gut–immune axis is mediated, at least in part, by short-chain fatty acids (SCFA) which are produced by the gut microbiome during the fermentation of dietary fiber [23]. These SCFAs interact with lymphocytes locally in the gut, as well as systemically [23]. SCFAs work through histone deacetylase inhibition to favor the Th1 phenotype, thereby promoting anticancer immunity (Figure 1) [24,25].

This review will explore the role of the gut microbiome and its interplay with UC and RCC, the mechanisms through which they interact, and current efforts to improve therapy and outcomes through modulation of the gut microbiome. This represents a novel understanding and approach to modulating the response to immunotherapy in immunogenic GU cancers.

## 2. Epidemiology and Therapeutics in UC and RCC

Bladder cancer is the ninth most common cancer worldwide, with an estimated 614,000 new cases and 220,000 deaths in 2022. UC constitutes approximately 90% of bladder cancer cases, with a high propensity for recurrence and progression [26]. RCC accounts for about 3% of adult malignancies, but remains among the deadliest urologic cancers, with clear cell RCC (ccRCC) comprising roughly 75% of RCC cases [27]. The predominant risk factor for UC is cigarette smoking, as well as environmental exposure; in RCC, smoking and obesity are risk factors, although they do not carry the same weight as smoking in UC [28,29].

The advent of ICIs targeting PD-1/PD-L1 and CTLA-4 axes has transformed the treatment of RCC and UC, with agents like nivolumab and ipilimumab, pembrolizumab, and avelumab being approved in various settings [4,5,30]. In RCC, CheckMate 214 demonstrated significant improvement with dual checkpoint inhibition with ipilimumab plus nivolumab compared with a single agent, TKI (sunitinib), with superior overall survival (OS) with median OS of 52.7 months vs. 37.8 month [4,13,31]. Additional studies evaluating combination of ICI and TKI include the following: KEYNOTE-426, which was combination of axitinib and pembrolizumab, OS 47.2mo vs. 40.8; CheckMate-9ER, which combined cabozantanib with nivolumab, OS 46.5 vs. 35.5; and KEYNOTE-B61, which combined Lenvatinib with pembrolizumab, OS 53.7 vs. 54.3, but with significantly longer PFS and duration of response [10,14,15,32,33,34]. Many have taken this to imply that, with appropriate immune modulation, durable responses to ICI are possible with dual checkpoint inhibition or TKI plus ICI. Dual ICI, in the form of ipilimumab plus nivolumab, also showed activity in non-clear cell RCC [35]. Subsequent to the integration of ICI into treatment for advanced disease, the PD-1 inhibitor pembrolizumab became the first adjuvant therapy for resected clear-cell RCC to show an overall survival benefit compared to placebo, with a HR of 0.66 (95% CI, 0.48−0.90) and estimated OS rate at 5 years of 87.7% vs. 82.3% [36,37].

The experience with UC and ICI is similar, initially with addition of ICI to first-line platinum-based chemotherapy, first in the form of maintenance ICI after chemotherapy with the PD-L1 inhibitor avelumab, and then later upfront addition of the PD-1 inhibitor nivolumab to first-line gemcitabine plus cisplatin [5,9]. Notably, doublet ICI without concurrent cytotoxic chemotherapy is not superior to chemotherapy in advanced UC, as demonstrated by the arm of CHECKMATE 901 that compared gemcitabine plus carboplatin to ipilimumab plus nivolumab [38]. However, the platinum chemotherapy plus ICI approach in a first-line metastatic setting was ultimately supplanted by ICI plus an antibody–drug conjugate based on findings from EV-302 [8]. In EV-302, the combination of pembrolizumab with the nectin-4 targeted antibody–drug conjugate enfortumab vedotin improved both PFS (hazard ratio [HR] 0.48; 95% CI, 0.41–0.57) and OS (HR 0.51; 95% CI, 0.43–0.61]), with an improvement in median PFS and OS of more than 6 and 17 months, respectively [39]. Subsequently, addition of PD-L1 blockade to perioperative chemotherapy was also shown to improve OS for patients with muscle-invasive bladder cancer undergoing cystectomy, wherein addition of pre- and post-operative durvalumab to neoadjuvant gemcitabine plus cisplatin achieved 24-month OS of 82.2% (95% CI, from 78.7 to 85.2) compared to 75.2% (95% CI, from 71.3 to 78.8) with chemotherapy alone (HR, 0.75; 95% CI, 0.59–0.93; *p* = 0.01). [7]. Adjuvant nivolumab after cystectomy also improved disease-free survival [40,41], and pembrolizumab was approved as an option for the treatment of BCG-unresponsive non-muscle-invasive bladder cancer with carcinoma in situ [42].

Based on these data, current guidelines for the management of advanced UC and RCC include the use of immunotherapy as first-line treatment in all patients who do not have a contraindication to immunotherapy [43,44]. While the implementation of immunotherapy into the treatment of GU cancers has improved outcomes, there is still a great deal that remains unknown with regard to predicting who will respond to therapy and how to improve response rates and achieve durable response. While current data suggest that tumor mutational burden (for UC), PD-L1 staining, IMDC risk score (for RCC), and certain histologic features may provide insights into ICI response rate, these data are not sufficiently accurate or precise to be used as predictive biomarkers in clinical practice [45]. This has stimulated interest in host factors such as the gut microbiome, which influences systemic immunity and may represent a modifiable determinant of immunotherapy response [18,19].

## 3. Microbiome and Carcinogenesis

There is emerging evidence that the gut microbiome is different between patients with cancer and those without. Diets which are high in processed foods and red meat and lower in fruits, vegetables, and whole grains are associated with increased risk of the development of GU cancers; one potential mechanism of this development is through the alteration of the gut microbiome [46,47]. One study of RCC patients prior to resection revealed differences in gut flora, with RCC patients having a higher abundance of Desulfovibrionaceae and less abundance of four kinds of lactobacillus [48]. In bladder cancer studies, there is some evidence to suggest that Bifidobacterium, Actinocabteria, and Ruminococcustorques are associated with increased risk and Allisonella was associated with decreased risk [49]. Rumicoccustorques and Erysipelatoclostridium were also linked to an increased risk of RCC [49]. While studies describing the associations between particular microbes with incidence of UC and RCC offer potential insights into the role the gut microbiome plays in carcinogenesis, it is not understood whether the cancer distorts the microbiome or whether a distorted microbiome increases the risk of cancer. It is also unknown whether or not mechanisms that modulate response to therapy also play a role in carcinogenesis.

One mechanism though which the gut microbiome interacts with the immune system is via the microbial production of metabolites, including the SCFAs acetate, butyrate, and propionate. SCFAs are one of the ways in which the gut microbiome interfaces with a number of different organ systems [50]. SCFAs from the gut can also modulate the urothelial mucosal barrier, and may play a protective role in tumorigenesis through suppression of inflammation [50].

## 4. Microbiome and Immune Interaction

The highest proportion of immunomodulatory bacteria are located in the distal small bowl and colon. These structures comprise an epithelial lining with intraepithelial cells, and mucin secreting cells which provide essential protective barrier [51]. Just deep to the epithelial cells is the lamina propria which is a connective tissue layer and home to a complex immune environment organized into Peyer’s patches. Within these Peyer’s patches, immune cells including T and B lymphocytes, natural killer cells, macrophages, dendritic cells and M cells [52]. These cells are continuously interacting with the commensal bacteria in the gut. Innate immune cells recognize pathogen-associated molecular patterns (PAMPs) and damage-associated molecular patterns (DAMPs) through toll-like receptors and nod-like receptors and interact with the adaptive immune system though cytokines [53]. Lymphocytes, located in Peyer’s patches, are essential for humoral immunity, including IgA production. They are also essential to maintaining homeostasis through a balance of pro- and anti-inflammatory mediators [54].

Bacterial components, such as lipopolysaccharides (LPS) and flagellin, engage pattern recognition receptors (PRRs) like Toll-like receptors on antigen-presenting cells, stimulating pro-inflammatory cytokines and chemokines [55]. This activation recruits CD8+ T cells and natural killer (NK) cells to the tumor microenvironment, enhancing immune surveillance. Dysbiosis with loss of such immunostimulatory microbes can impair these processes [56]. While ICI targets cytotoxic T lymphocytes (CTLs) and there is much focus on optimizing the lymphoid compartment for therapeutic benefit, there is a large body of evidence linking myeloid-derived suppressor cells (MDSC) as pro-tumorigenic [57]. In gastrointestinal cancer models, gut microbiome dysbiosis has been shown to increase MDSCs and drive tumor progression [58].

SCFAs are produced by the gut microbiome through fermentation of fiber (Figure 1). These SCFAs exert divers immunomodulatory effects, including influencing anti-tumor immunity [18,59]. One such mechanism is through epigenetic modulation. Butyrate functions as a histone deacetylase (HDAC) inhibitor that facilitates histone acetylation and upregulation of CTL through increase in IFN-ƴ and granzyme B and skew TC17 cells to a CTL phenotype [60,61].

In addition to HDAC inhibition, butyrate has been shown to modulate the inflammatory response in vivo, and in vitro studies in mice have demonstrated that butyrate induces T regulatory cells, which decrease inflammation [62]. Butyrate also plays a role in cytokine milieu, which favors Th1 differentiation and thereby promotes anti-tumor immunity [25]. In melanoma, patients that were responders to immunotherapy had a higher abundance of Ruminococcaceae/*Faecalibacterium* and a more diverse microbiome than non-responders, who had a higher abundance of bacteroidales. Ruminococcaceae/*Faecalibacterium*—which are known to produce high levels of butyrate—improve antigen presentation, T cell function, and lymphoid penetration into the tumor microenvironment compared to those with less diverse gut microbiota and higher abundance of Bacteroidales [18].

In RCC, SCFA-producing bacteria like *Akkermansia muciniphila* and *Bacteroides saly ersiae* are associated with favorable immune profiles and improved ICI response. Moreover, introduction of these beneficial commensals into mouse models of RCC improved ICI response in vivo [20].

In addition to SCFA, other metabolites, including bile acids, tryptophan derivatives, and polyamines, are generated or modified by the gut microbiome, and these microbial metabolites play a role in modulating anti-tumor immunity [63,64,65]. Gut bacteria convert primary bile acids into secondary bile acids, produce tryptophan-derived indoles, and synthesize polyamines from dietary and host substrates [66,67]. These metabolites act as signaling molecules that influence the differentiation, activation, and function of immune cells within the tumor microenvironment [63,68]. For example, microbiota-transformed bile acids and tryptophan metabolites can promote immunosuppressive phenotypes in regulatory T cells and myeloid-derived suppressor cells, while polyamines can impair anti-tumor T cell responses and foster immune evasion. Dysregulation of the tryptophan metabolic pathway is associated with a pro-tumorigenic immunosuppressive microenvironment and resistance to immunotherapy (Figure 1) [69,70,71]. 

While the gut microbiome’s role as a modulator of immunotherapy has been extensively studied, the influence of the urinary microbiome remains relatively unexplored. Exogenous microbes can clearly impact UC when introduced to the bladder, with the most well-known example being BCG, a live mycobacteria that is a highly effective intravesical therapy for early stage bladder cancer [72]. In two out of three recent randomized clinical trials testing the combination of BCG with ICI for BCG-naïve non-muscle-invasive bladder cancer, the combination of BCG with ICI was shown to improve event-free survival compared to BCG [73,74,75]. However, whether the improvement in event-free survival was attributable to a synergy between BCG and ICI versus simply additive efficacy remains unknown.

UC also disrupts the urothelial mucosa, resulting in urine microbiome dysbiosis and an increased intravesical abundance of pathogenic bacteria such as *Pseudomonas*, *Acinetobacter*, *Micrococcus*, *Sphingomonas*, and *Ralstonia* [76]. Such bacteria can trigger local and systemic inflammation, as indicated by increased CRP and neutrophil-to-lymphocyte ratio in the presence of positive urine cultures [77]. Unlike BCG, which is an attenuated bacteria easily controlled by the immune system, the pathogenic bacteria associated with UC can overcome the immune response and trigger uncontrolled inflammation, potentially leading to immune cell exhaustion that might impair an anti-tumor immune response [78].

In summary, the gut microbiome interacts with the immune system locally in the gut and systemically by modulating both the lymphoid and myeloid compartments. SCFAs shift T cells to a Th1 phenotype increasing INF-ƴ through HDAC inhibition. The role of the urinary microbiome and its interaction with the local and systemic immune system is much less clear, and additional studies are needed to further explore this relationship.

## 5. Associations of the Microbiome and ICI Efficacy

Recognizing that the gut plays a crucial role in immune modulation, many studies have sought to identify the aspects of the gut microbiome that influence response to ICI. The precise ecosystem of bacteria to elicit ideal anti-tumor immunity is not well elucidated, although preclinical studies suggest that bifidobacterium and *Akkermansia muciniphila* are favorable bacteria and that supplementation with these bacteria can restore the anti-tumor immune response in non-responding mice [19,21]. Additionally, *Bacteroides thetaiotaomicron* and *Bacteroides fragilis* have been implicated in enhancing ICI efficacy through an IL-12-dependent manner similar to what was shown with *A. muciniphila* [19,79,80]. Multiple studies have used microbiome sequencing to characterize the gut microbiome of patients with GU cancers. Bifidobacterium, actinobacteria, and ruminococcustorues were seen in higher proportions in bladder cancer patients compared to the healthy control. *Faecalibacterium prausnitzii* and *Akkermansia muciniphila allisonella* and *oscillbacer* were associated with a lower risk of GU cancers and improved outcomes [20,49,81,82]. Derosa et. al. profiled the gut microbiome of 69 metastatic RCC patients treated with nivolumab; they found that patients with a higher abundance of *Faecalibacterium prausnitzii* and *Akkermansia muciniphila* had improved outcomes [20]. In the PURE-01 trial, a study of neoadjuvant pembrolizumab for patients with muscle-invasive UC, the Sutterella genus appeared to be enriched among ICI responders, while *Ruminococcus bromii* was enriched among non-responders [83]. Treatment with antibiotics prior to ICI reduced response rates to nivolumab [20]. TKIs also had an influence on the gut microbiome and shifted the species present; however, there was no distinct correlation with outcome [20]. TKI’s side effect profile is also influenced by the gut microbiome. In a small study of 20 RCC patients treated with VEGF-TKI, diarrhea was more common in patients with higher levels of *bacteroides* and *prevotella* and lower levels of *bifidobacterium* [84]. Chemotherapy is also frequently used in early-stage bladder cancer, and preclinical evidence suggests that the gut microbiome influences metabolism of chemotherapy through immune modulation [85].

Given variable findings across studies, prediction of response to immunotherapy based on the gut microbiome continues to be a difficult dilemma. While different bacterial taxa have been implicated, a recent study published the TOPOSCORE, which uses a 21 gene PCR to create a personalized score that can predict response to immunotherapy. This was validated across hundreds of patients in both NSCLC and GU cancers [86]. Prediction models can inform patients about risk and help guide therapeutic interventions. This score is an attempt to standardize microbiome abundance analyses, as there are many different techniques which yield substantially different results depending on the technique used [87].

Understanding which bacteria lead to more favorable outcomes may help to guide therapeutic efforts. However, looking further downstream at microbiome-derived metabolites, including SCFAs, is also a promising approach, as these may mediate the influence of the gut microbiome on anti-tumor response. For example, in a cohort study of 52 patients receiving ICI for a variety of solid tumors, high fecal and plasma concentrations of select SCFAs were associated with longer PFS, including fecal acetic acid (HR, 0.29; 95% CI, 0.15–0.54), propionic acid (HR, 0.08; 95% CI, 0.03–0.20), butyric acid (HR, 0.31; 95% CI, 0.16–0.60), valeric acid (HR, 0.53; 95% CI, 0.29–0.98), and plasma isovaleric acid (HR, 0.38; 95% CI, 0.14–0.99) [88]. Among patients receiving CAR-T therapy for non-Hodgkin lymphoma, patients with lower serum butyrate levels have poorer survival, implying the importance of butyrate for optimal T cell function [89]. This has also been demonstrated in NSCLC, where patients with higher levels of SCFAs (acetic acid, propionic acid, and butyric acid) had higher response rates to ICI than those with lower levels [90].

In melanoma and pancreatic tumor models, the SCFAs pentanoate and butyrate have been shown to enhance the anti-tumor effects of CTLs and Chimeric antigen receptor T (CAR-T) through upregulation of mTOR signaling and HDAC inhibition. This leads to increased production of the anti-tumor mediators CD25, INF-ƴ, and TNF-α [91]. Another study demonstrated that in vitro and in vivo supplementation with the SCFA butyrate enhanced the frequency of CTL and stimulated effector memory cells with high expression of IL-15Rβ and T-bet [60]. CAR-T, when exposed to butyrate in vitro, has upregulated specific lysis and a higher expression of activation markers and costimulatory molecules [89]. SCFAs measured in serum have been correlated with outcomes in patients receiving CART therapy for non-Hodgkin’s lymphoma. While CAR-T is not yet a standard treatment option for either UC or RCC, emerging evidence indicates promising potential for CAR-T as a novel immunotherapy for RCC and UC [92,93,94].

Additionally, butyrate may play a role in the efficacy of conventional chemotherapy [95]. In a pancreatic cancer mouse model, butyrate improved the efficacy of gemcitabine by reducing the stromatogenesis of the cancer, and also reduced toxicity [95]. Gemcitabine is frequently used in both localized and advanced urothelial cancer, and optimization of gemcitabine efficacy could have impacts on UC outcomes at these various stages of the disease [44].

## 6. Antibiotics and Dysbiosis

It is well established that antibiotics can disrupt the normal gut flora, and multiple studies have reported associations between antibiotic exposure and poor outcomes on ICI, including among patients with RCC and UC [20,96]. Antibiotic-induced dysbiosis also serves as a possible barrier to outcomes in patients treated with immunotherapy [97]. In patients receiving CAR-T therapy, broad-spectrum antibiotics are associated with worse outcomes [89]. In over 200 pts with RCC treated with ICI, OS was worse in patients with lower levels of soluble MAdCAM-1, which retains T_reg_17 cells in the gut, thereby reducing their migration to the tumor microenvironment. Low-serum-soluble MAdCAM-1 was identified as a proxy of intestinal dysbiosis and a robust predictor of shorter PFS and OS of RCC, UC, and lung cancer patients receiving ICI [98]. Additional studies support worse outcomes and response rates to ICI when patients were treated with broad-spectrum antibiotics, including a large-scale meta-analysis with over 40,000 patients, including 28% who had received broad-spectrum antibiotics. In the meta-analysis, the pooled HRs associated with antibiotic use for OS and PFS were 1.61 (95% CI, 1.48–1.76) and 1.45 (95% CI, 1.32–1.60) [99]. The indication for antibiotics is not frequently documented, and patients requiring antibiotics may have worse outcomes due to additional confounders such as co-morbidities or malignancy-associated complications.

## 7. Therapeutic Approaches to Modifying the Microbiome to Enhance ICI

There have been three landmark interventional trials in RCC which have paved the way for additional large-scale studies which modulate the gut microbiome to augment response to anticancer therapy, namely ICI in GU cancers, which are summarized in Table 1. Additional details on these trials, as well as other potential approaches to microbiome modification, are summarized below.

### 7.1. FMT

While prebiotics can alter the microbiome, they are somewhat limited in composition, and it is not clear how they will interact with the existing microbiome. FMT offers a more comprehensive approach, and can replace a potentially protumorigenic microbiome. In preclinical models, germ-free mice have impaired anti-tumor immune response to ICI compared with mice with intact gut microbiome. This was also replicated in mice who were given antibiotics that target the microbiome and disrupt the normal function [19]. When germ-free mice were given a fecal microbiota transplant (FMT) from patients who responded to ICI therapy, the anti-tumor effect of ICI was restored; in contrast, FMT from non-responder patients did not restore ICI benefit [19]. Using an immunogenic tumor model of melanoma, mice with spontaneous anti-tumor immunity who were housed with non-responding mice or given non-responding mouse FMT lost the anti-tumor effect of ICI [21]. The TACITO trial, presented at ESMO 2024, was a phase 2 study randomizing metastatic RCC patients to axitinib + pembrolizumab with or without FMT [102]. Patients received colonic infusion at baseline and then at oral FMT 90 and 180 days after starting immunotherapy. PFS at 1 year was 66.7% in the FMT group and 35% in the placebo group [102]. Limitations of this study include a single donor for the FMT. The donor was chosen due to history of metastatic RCC with complete response to immunotherapy that had been durable for years. However, FMT is relatively invasive, and is also limited in its utility by the availability of suitable donors. Notably, FMT intervention has also demonstrated promising activity in clinical trials for non-GU malignancies, including treatment-naïve non-small-cell lung cancer, as well as treatment-naïve and ICI-refractory melanoma [103,104,105,106], although activity is much less certain for gastrointestinal malignancies based on the available data [107,108].

### 7.2. Probiotics

Probiotics—which are live, natural microorganisms administered orally—offer an exciting potential therapeutic approach, as they can be implemented on a large scale with relative ease of administration and standardization of dosing [109]. Two small, randomized phase 1 trials, each consisting of approximately 30 patients with advanced RCC, suggested significant efficacy of CBM588 (*Clostridium butyricum*), a probiotic initially used in Japan for antibiotic-induced diarrhea. Compared to ICI alone, addition of CBM588 to dual ICI with ipilimumab and nivolumab improved PFS, with a median PFS of 12.7 months with CBM588 versus 2.5 months with ICI alone (HR 0.15; 95% CI 0.05–0.47, *p*  =  0.001). In the second randomized study, CBM588 was added to the ICI + TKI combination of nivolumab plus cabozantanib and improved ORR to 74% versus 20% without CBM588 (*p*  =  0.01) [100,101]. It has been noted that the control arms in these trials fared poorly compared to historical controls for unclear reasons. Nonetheless, there is optimism regarding tentative plans for a larger phase 3 trial for RCC called SWOG BioFront, which intends to confirm or refute CBM588 as a potential therapeutic for ICI enhancement [110].

Notably, other data have indicated potentially concerning results for probiotics, including a phase 1b trial of SER-401, a spore-based formulation enriched with *Firmicutes*, although the 14-patient trial was too small to make definitive conclusions [111]. The trial randomly assigned 14 patients with advanced melanoma who were administered nivolumab to SER-401 or placebo groups, and showed an objective response rate of 25% in the SER-401 arm and 67% in the placebo arm. Correlative analyses suggested that this potentially detrimental effect may have been attributable to gut dysbiosis induced by the vancomycin used for gut preconditioning in the SER-401 arm—a potentially salient lesson about the risks of antibiotic-induced dysbiosis for future clinical trials.

### 7.3. Diet and Prebiotics

Patient diet is known to influence the gut microbiome [112]. In particular, prebiotics like dietary fiber (e.g., inulin, pectin, resistant starch) are fermented by gut bacteria into SCFAs, which modulate the immune system [113]. Observational studies demonstrate that patients consuming high-fiber diets may have improved outcomes when treated with ICI compared to those with low-fiber diets [114], with emerging data supporting this association in genitourinary cancers as well [115]. The Mediterranean diet has also been associated with favorable outcomes in patients with advanced melanoma on ICI [116]. Diet interventions are scalable, and can be implemented broadly with low costs on the health care system and patients; however, proving causation in observational studies on diet and nutrition is difficult. Large interventional and ideally randomized trials are clearly needed to confirm causality. However, such trials can be challenging to execute, especially regarding controlling participant dietary and probiotic intake. Based on preliminary observational data, there is an ongoing clinical trial in stage III–IV melanoma, referred to as the DIET study. The DIET study randomized patients on ICI to either a high-fiber diet (30–50 g/day of fiber) or a standard healthy diet (20 g/day of fiber), using prepared meals for each arm for up to 11 weeks [117]. The study’s primary objective was to establish the effects of the high-fiber diet on the structure and function of the gut microbiome. However, exploratory analysis of cancer-specific outcomes from the DIET study (*n* = 45) showed that, in the combined neoadjuvant and unresectable cohort (*n* = 24), the objective response rate in the high-fiber group was 77% compared to 29% with the control diet (*p* = 0.06) [118]. Median follow-up at time of reporting was 22.6 months. In the neoadjuvant cohort (*n* = 12), median event-free survival was not reached in the high-fiber group versus 20 months in the control group (*p* = 0.03), although the pathological complete response rate was similar, being 57% vs. 50%, respectively (*p* = 1.0). In the adjuvant group (*n* = 19), which had a median follow-up of 27.6 months, the recurrence rate was 14% with the high-fiber diet and 33% with the control diet (*p* = 0.56). Grade 3 and higher immune-related adverse events were numerically lower with the high-fiber diet, at 28.6% with the high-fiber intervention vs. 40.0% with the control diet (*p* = 0.51). While most of these differences were not statistically significant, the numerical trends consistently favoring the high-fiber intervention suggest that larger randomized trials with longer follow-up are warranted to more definitively assess whether dietary intervention targeting the gut microbiome can serve as an adjunct to anticancer therapy to augment response and improve outcomes.

Another prebiotic of interest is camu-camu, a polyphenol-rich berry which, in preclinical models, enriched the abundance of the gut bacteria associated with ICI response (*Ruminococcaceae* and *Alistipes*) and in two patients with ICI-refractory melanoma, camu-camu plus anti-PD-1 rechallenge resulted in deep clinical responses [119,120]. Pre-clinical models also suggest that a ketogenic diet might enhance ICI-induced anti-tumor immune response. Ketogenic diet was also associated shifts in the gut microbiome to species such as *Eisenbergiella massiliensis* that correlated with high levels of the ketone body 3-hydroxybutyrate [121].

## 8. Conclusions and Future Perspectives

UC and RCC treatment has been revolutionized by the advent of immunotherapy, in the form of immune checkpoint blockade. However there remains a significant portion of patients who have limited duration of response or who do not respond at all [31,39]. The gut microbiome is a diverse ecosystem that plays an important role in immune cell education and function and poses a unique target for modulation of immunotherapy response; the potential modulation and mechanisms of immune cell alteration are highlighted in Figure 1. The gut microbiome has the potential to serve as an additional biomarker/prognostic tool to help provide guidance to patients regarding prognosis and response to therapy. This highlights the importance of additional studies with consistent methods.

Modulation of microbiomes represents an exciting potential opportunity to enhance oncologic therapies, but the current literature has notable limitations. While preclinical data and observational clinical studies provide compelling preliminary evidence, large-scale randomized clinical trials of microbiome-based interventions are still needed to verify a causal link between the microbiome and ICI efficacy before microbiome-based interventions can be integrated into standard clinical practice. The few clinical trials which are randomized are small and necessitate replication in larger studies to confirm their findings [100,101]. Methods used to process stool specimens and determine bacterial abundance often vary between studies, and there is not a standardized method for fecal microbiome analysis. This was highlighted by Nearing et al., who analyzed 14 different abundance-testing methods and produced vastly different results from 16S rRNA gene datasets [87]. Ultimately, standardization of microbiome profiling and data normalization, as well as integration of microbiome metrics with other predictive biomarkers such as tumor mutational burden, are needed to enable the transition of microbiome research from the bench to the bedside.

Table 2 outlines some of the cutting-edge ongoing clinical trials on GU cancers that target the microbiome. These will provide additional insights into the efficacy and feasibility of using the microbiome–immune axis in GU oncology. With our continued understanding of the interplay between the immune system, gut microbiome, and GU malignancies, further large-scale therapeutic trials are needed to demonstrate the most effective and feasible avenues to target the gut microbiome to improve patient outcomes.

## Figures and Tables

**Figure 1 cancers-17-03606-f001:**
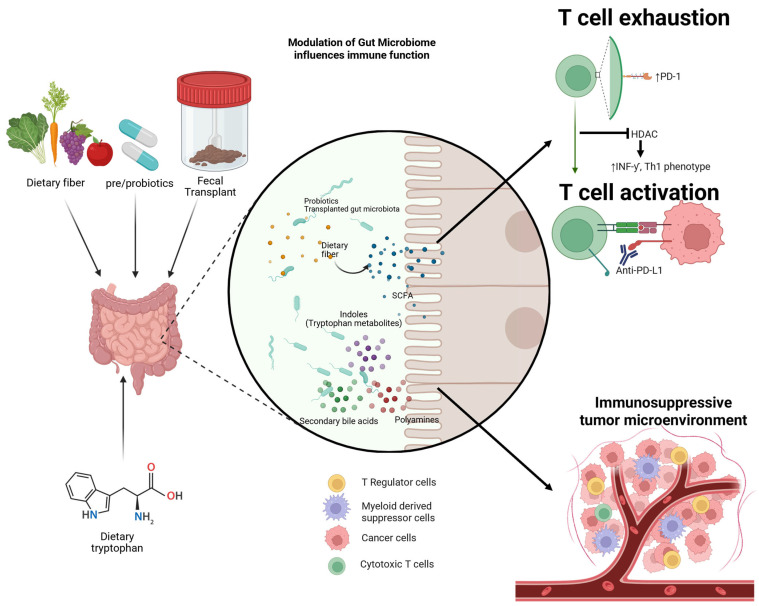
Modulation of gut microbiome influences immune function. Visual representation of microbiome-targeting interventions and the proposed mechanism of improved immune function, and the mechanism through which metabolites play an immunosuppressive role in the tumor microenvironment.

**Table 1 cancers-17-03606-t001:** Landmark clinical trials for gut-microbiome-directed intervention in genitourinary oncology.

	Nivolumab Plus Ipilimumab with or Without Bacterial Supplementation in Metastatic Renal Carcinoma: A Randomized Phase 1 Trial	Cabozantinib and Nivolumab with or Without Live Bacterial Supplementation in Metastatic Renal Cell Carcinoma: A Randomized Phase 1 Trial	LBA77 Fecal Microbiota Transplantation (FMT) Versus Placebo in Patients Receiving Pembrolizumab Plus Axitinib for Metastatic Renal Cell Carcinoma: Preliminary Results of the Randomized Phase 2 TACITO Trial
ClinicalTrials.gov Identifier	NCT03829111	NCT05122546	NCT04758507
Study Citation	Dizman, N., Meza, L., Bergerot, P. et al. *Nat Med* **28**, 704–712 (2022). [100]	Ebrahimi H, Dizman N, Meza L, et al. *Nat Med*. 2024;30(9):2576–2585. [101]	Ciccarese, C. et al. LBA77, *Annals of Oncology*, Volume 35, Supplement 2, 2024, Page S1264, [102]
Patient Population	*n* = 29 Metastatic renal cell carcinoma (mRCC), treatment-native	*n* = 30 Advanced or metastatic renal cell carcinoma (mRCC), treatment native	*n* = 50 Metastatic renal cell carcinoma
Treatment Regimen	Nivolumab + ipilimumab	Cabozantinib + Nivolumab	Axitinib + Pembrolizumab
Microbiome Intervention	Probiotic CBM588 (*Clostridium butyricum)*	Probiotic CBM588 (*Clostridium butyricum*)	FMT
Primary Endpoint	Effect of CBM588 on relative abundance of gut microbial populations and specifically *Bifidobacterium* spp.	Change in the relative abundance of *Bifidobacterium* spp.	Increase of ≥ 20% in the rate of patients with no-disease progression at 1-year (1-year PFS) with FMT vs. without
Result of primary analysis	Negative No significant from baseline to week 12 in experimental arm or control (*p* = 0.304, *p* = 0.461)	Negative No significant from baseline to week 13 in experimental arm or control (*p* = 0.95, *p* = 0.39)	Positive Axitinib + Pembrolizumab with FMT (66.7% vs. 35%, *p* = 0.036)
Key Secondary Clinical Efficacy Endpoints	Comparing experimental arm (nivolumab + ipilimumab + CBM588) to control (nivolumab + ipilimumab): PFS (12.7 vs. 2.5 months; HR 0.15, 95% CI 0.05–0.47, *p* = 0.001) ORR (58% vs. 20%, *p* = 0.06) Reduction in tumor target lesions (74% vs. 50%) Disease control (79% vs. 40%)	Comparing experimental arm to control: PFS at 6 months (84% vs. 60%) ORR (74% vs. 20%, *p* = 0.01) Reduction in tumor target lesions (89% vs. 80%)	Comparing experimental arm to control: Median PFS was 14.2 months (95% CI, 0.9–27.6) vs. 9.2 months (95% CI, 3.0–15.4) ORR (54% vs. 28%)

FMT, fecal microbiota transplant; ORR, objective response rate; PFS, progression-free survival

**Table 2 cancers-17-03606-t002:** Ongoing interventional and observational trials evaluating gut microbiome in GU cancers.

Therapy	Phase	Patient Population	Trial Identifier	Status
**Interventional Trials**				
**Probiotic/Prebiotic**				
Cabozantinib + Nivolumab with CBM588	1	Advanced or metastatic renal cell carcinoma (mRCC)	NCT05122546	Active, not recruiting
Pembrolizumab with CBM588	2	RCC T2-4 any grade, N0M0; T TxNxM0-1	NCT07037004	Not yet recruiting
Nivolumab + ipilimumab with CBM588	1	Advanced or metastatic RCC	NCT06399419	Recruiting
Nivolumab with BMC128	1	mRCC or clear cell renal cell carcinoma (ccRCC)	NCT05354102	Active, not recruiting
**Fecal Microbiota Transplant**				
Nivolumab + ipilimumab with FMT (PERFORM)	1	Advanced or mRCC	NCT04163289	Active, not recruiting
**Dietary Interventions**				
Nivolumab + ipilimumab or single agent ipilimumab/nivolumab/pembrolizumab with ketogenic diet	1	Metastatic melanoma or mRCC	NCT06391099	Recruiting
Nivolumab + ipilimumab/relatimab or single agent ipilimumab/nivolumab/pembrolizumab with ketogenic diet	1/2	Melanoma or ccRCC or mRCC	NCT06896552	Not yet recruiting
Nivolumab + ipilimumab with inulin gel	1/2	Advanced or mRCC	NCT06866262	Recruiting
**Observational Trials**				
ICIs (single agent or combination) effect on gut microbiota (PARADIGM)		NSCLC, malignant melanoma, RCC, TNBC; any stage	NCT05037825	Recruiting
Investigate how the microbiome correlates with efficacy and toxicity of ICIs) in patients with advanced cancer		Advanced melanoma, RCC, or NSCLC	NCT04107168	Unknown Status
Establish the microbiota composition as a predictive tool for the response to the intravesical immunotherapy with BCG or Gem/Dox or MMC		Non-muscle invasive bladder cancer (NMIBC)	NCT06675656	Not yet recruiting
Investigating differences in the bladder microbiome in urothelial cacinoma		Urothelial Carcinoma	NCT06992986	Recruiting
Microbiota profiling in urine and bladder tissue of male healthy individuals and patients with bladder cancer		Bladder Cancer	NCT06289283	Active, not recruiting

ICI, immune checkpoint inhibitor; NSCLC, non-small-cell lung carcinoma; TNBC, triple-negative breast cancer; BCG, Bacillus Calmette-Guérin; MMC, Mitomycin-C.

## Data Availability

Data are contained within the article.

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
