# Peer review of "The Microbiome and Genitourinary Cancers: A New Frontier"

_cancers, 2025, doi:10.3390/cancers17223606_

Round 1
Reviewer 1 Report
Comments and Suggestions for Authors
This review provides a timely and comprehensive synthesis of the rapidly evolving role of the gut microbiome in shaping the efficacy of immunotherapy for genitourinary cancers, notably urothelial and renal cell carcinoma. It effectively bridges basic science—detailing mechanisms like microbial metabolite-driven immune modulation—with emerging clinical evidence, highlighting innovative therapeutic strategies such as fecal microbiota transplantation and probiotics. The manuscript stands out by critically evaluating the potential of microbiome modulation to overcome treatment resistance, thereby offering a compelling perspective on a promising frontier in oncology with significant implications for future research and clinical practice.
#1 Sentence "Introduction of immune checkpoint inhibitors (ICIs)-monoclonal antibodies that bind and block the immune checkpoint pathway proteins PD-1, PDL-1 or CTLA-4-has resulted in significant improvement in the survival of patients with UC and RCC. checkpoint pathway proteins PD-1, PDL-1 or CTLA-4-has resulted in significant improvement in the survival of patients with UC and RCC, either alone or in combination with other agents such as anti-body-cells. either alone or in combination with other agents such as anti-body-drug conjugates or VEGF tyrosine kinase inhibitors" lacks key references (PMID: 38652128). The authors add citations to the above key references, which will provide important reference value for subsequent studies.
#2 The introduction of microbiome-immunotherapy linkage in the background is a bit brief, and fails to fully elaborate the theoretical basis of the “gut-immunity axis” in cancer; add a paragraph at the end of the introduction to explicitly state the core hypothesis that “the gut microbiome influences the response to cancer therapy by modulating systemic immunity”, and add a paragraph to clearly state that “the gut microbiome influences the response to cancer therapy by modulating systemic immunity”. The core hypothesis of “the gut microbiome influences cancer therapeutic responses by modulating systemic immunity” is explicitly stated in a paragraph at the end of the introduction, and key mechanisms such as short-chain fatty acids and T-cell polarization are briefly mentioned.
#3 Mechanisms by which the microbiome influences immunity (Section 4) is fragmented and lacks systematic summarization; add a summary or diagram in Section 4 to systematically summarize the mechanisms by which the microbiome influences the tumor microenvironment through “metabolites (e.g., SCFAs) - immune cells - cytokines” and “tumor microenvironment”. The pathway through which the microbiome influences the ICI response through "metabolites (e.g. SCFAs)-immune cells-cytokines-tumor microenvironment.
Given these considerations, I highly recommend that authors revise their manuscript. Looking forward to receiving your revised version of the manuscript. I will review this manuscript again based on the revised version.
Author Response
Reviewer 1:
Comment 1:
#1 Sentence "Introduction of immune checkpoint inhibitors (ICIs)-monoclonal antibodies that bind and block the immune checkpoint pathway proteins PD-1, PDL-1 or CTLA-4-has resulted in significant improvement in the survival of patients with UC and RCC. checkpoint pathway proteins PD-1, PDL-1 or CTLA-4-has resulted in significant improvement in the survival of patients with UC and RCC, either alone or in combination with other agents such as anti-body-cells. either alone or in combination with other agents such as anti-body-drug conjugates or VEGF tyrosine kinase inhibitors" lacks key references (PMID: 38652128). The authors add citations to the above key references, which will provide important reference value for subsequent studies.
Response 1:
We agree with the additional reference and it has been added to the manuscript. This has been added on page 2 paragraph 1, reference number 11.
Comment 2:
#2 The introduction of microbiome-immunotherapy linkage in the background is a bit brief, and fails to fully elaborate the theoretical basis of the “gut-immunity axis” in cancer; add a paragraph at the end of the introduction to explicitly state the core hypothesis that “the gut microbiome influences the response to cancer therapy by modulating systemic immunity”, and add a paragraph to clearly state that “the gut microbiome influences the response to cancer therapy by modulating systemic immunity”. The core hypothesis of “the gut microbiome influences cancer therapeutic responses by modulating systemic immunity” is explicitly stated in a paragraph at the end of the introduction, and key mechanisms such as short-chain fatty acids and T-cell polarization are briefly mentioned.
Response 2:
We agree and have added additional paragraphs in the introduction which further elaborate on the mechanism for modulation of the immune response by the microbiome.
Comment 3:
#3 Mechanisms by which the microbiome influences immunity (Section 4) is fragmented and lacks systematic summarization; add a summary or diagram in Section 4 to systematically summarize the mechanisms by which the microbiome influences the tumor microenvironment through “metabolites (e.g., SCFAs) - immune cells - cytokines” and “tumor microenvironment”. The pathway through which the microbiome influences the ICI response through "metabolites (e.g. SCFAs)-immune cells-cytokines-tumor microenvironment.
Response 3:
We agree and have added summary statement at conclusion of section 4 as well as additions to Figure 1 to depict mechanisms of action more clearly.
Reviewer 2 Report
Comments and Suggestions for Authors
With great interest, I reviewed the manuscript entitled “The Microbiome and Genitourinary Cancers: A New Frontier.” The authors present a timely and comprehensive review exploring the interplay between the microbiome and genitourinary (GU) cancers. This is a clinically and biologically relevant topic. The review demonstrates extensive literature coverage and genuine engagement with the field. However, in its current form, the manuscript remains primarily descriptive and would benefit from a more critical, structured, and conceptually integrated approach.
Major comments:
Conceptual focus and novelty: The paper summarizes a wide range of studies but lacks a clearly defined conceptual framework. The authors should clarify what constitutes the “new frontier” — whether mechanistic understanding, translational perspective, or therapeutic application. A graphical model linking microbiome alterations with immune pathways in GU cancers could strengthen the narrative.
Balance between gut and urinary microbiome: The review focuses extensively on the gut microbiome, while the urinary microbiome is addressed only briefly. A more detailed discussion of urinary microbial signatures in bladder cancer and their implications for local immune responses is warranted for a balanced GU perspective.
Mechanistic interpretation: The manuscript discusses short-chain fatty acids (SCFAs) effectively but should also integrate other metabolites such as bile acids, tryptophan derivatives, and polyamines — all known to modulate tumor immunity. Their connection to immune checkpoint inhibitor (ICI) response and resistance deserves elaboration.
Critical appraisal of evidence: Many cited studies are small or exploratory. The authors should explicitly discuss study limitations (sample size, dietary and antibiotic confounders, sequencing variability) and the current level of evidence supporting causality. A short subsection summarizing methodological challenges in microbiome studies would improve rigor.
Structure and flow: The manuscript would benefit from smoother transitions between mechanistic, clinical, and translational sections. Some paragraphs read as disconnected summaries. Consider using brief linking sentences or schematic figures showing how microbial modulation affects systemic and local immune dynamics.
Conclusions and future perspectives: The concluding section should go beyond summarizing. The authors should outline a concise roadmap for future research:
-
- standardization of microbiome profiling and data normalization,
- integration of microbiome metrics with immune and genomic biomarkers (PD-L1, TMB),
- and clinical trials assessing microbiome-targeted interventions (dietary modulation, probiotics, FMT).
Minor Comments:
- The Abstract is overly descriptive; condense and emphasize novelty and clinical implications.
- Keywords: add “immunotherapy resistance,” “urinary microbiome,” and “short-chain fatty acids.”
- Figures and Tables: simplify Table 2 to focus on GU-related studies and clarify legends for clinical relevance.
- References: check uniform formatting, capitalization, and inclusion of DOIs.
- Correct minor typographical and grammatical inconsistencies throughout the text.
Comments on the Quality of English Language
The English language is understandable but requires extensive polishing to improve precision and remove redundancy. Frequent use of “additionally” and “furthermore” can be reduced. A professional language edit is strongly advised.
Author Response
Reviewer 2:
Comment 1: Conceptual focus and novelty: The paper summarizes a wide range of studies but lacks a clearly defined conceptual framework. The authors should clarify what constitutes the “new frontier” — whether mechanistic understanding, translational perspective, or therapeutic application. A graphical model linking microbiome alterations with immune pathways in GU cancers could strengthen the narrative.
Response 1:
In order to clarify what constitutes a “new frontier”, we now explained in the abstract that “The microbiome plays a complex and dynamic role in regulating the immune system and represents a new frontier as a promising target for modulating response to immunotherapy.” We have updated Figure 1 to describe the mechanisms for microbiome modulation, including its impact on T cell function and anti-cancer immunity. We have added additional language to the introduction to better frame the review.
Comment 2:
Balance between gut and urinary microbiome: The review focuses extensively on the gut microbiome, while the urinary microbiome is addressed only briefly. A more detailed discussion of urinary microbial signatures in bladder cancer and their implications for local immune responses is warranted for a balanced GU perspective
Response 2:
We agree and while there is less that is known about the urinary microbiome and how this relates to GU cancer response to therapy, we have added paragraphs to the end of section 4 which provide additional detail regarding the urinary microbiome.
Comment 3:
Mechanistic interpretation: The manuscript discusses short-chain fatty acids (SCFAs) effectively but should also integrate other metabolites such as bile acids, tryptophan derivatives, and polyamines — all known to modulate tumor immunity. Their connection to immune checkpoint inhibitor (ICI) response and resistance deserves elaboration.
Response 3:
We agree and have included additional information about additional microbiome metabolites and their impact on the immune system and anti-cancer immunity in section 4, and incorporated them into figure 1.
Comment 4:
Critical appraisal of evidence: Many cited studies are small or exploratory. The authors should explicitly discuss study limitations (sample size, dietary and antibiotic confounders, sequencing variability) and the current level of evidence supporting causality. A short subsection summarizing methodological challenges in microbiome studies would improve rigor.
Response 4:
Thank you for this comment as it is important to recognize that the majority of clinical studies are small and there is not an accepted standard for microbiome profiling. We have added more critiques of the data in the conclusion section of the text.
Comment 5:
Structure and flow: The manuscript would benefit from smoother transitions between mechanistic, clinical, and translational sections. Some paragraphs read as disconnected summaries. Consider using brief linking sentences or schematic figures showing how microbial modulation affects systemic and local immune dynamics.
Response 5:
Thank you we have added additional text to help transition between topics to help make this review flow more smoothly and make it more readable figure 1 has also been modified to be a more comprehensive schematic representation of gut- immune system interplay.
Comment 6:
Conclusions and future perspectives: The concluding section should go beyond summarizing. The authors should outline a concise roadmap for future research:
- standardization of microbiome profiling and data normalization,
- integration of microbiome metrics with immune and genomic biomarkers (PD-L1, TMB),
- and clinical trials assessing microbiome-targeted interventions (dietary modulation, probiotics, FMT).
Response 6:
Thank you, this is a helpful structure for the conclusion section, it has been restructured with additional critiques added.
Comment 7:
Minor Comments:
- The Abstract is overly descriptive; condense and emphasize novelty and clinical implications.
- Keywords: add “immunotherapy resistance,” “urinary microbiome,” and “short-chain fatty acids.”
- Figures and Tables: simplify Table 2 to focus on GU-related studies and clarify legends for clinical relevance.
- References: check uniform formatting, capitalization, and inclusion of DOIs.
- Correct minor typographical and grammatical inconsistencies throughout the text.
Response 7:
- Abstract has been edited to be more concise
- Key words were added
- Table 2 was edited to clarify relevance
- References checked were all subjected to same format as requested by the journal
- Texted has been edited for typographical and grammatical errors.
Reviewer 3 Report
Comments and Suggestions for Authors
Your manuscript focuses on the treatment of genitourinary cancers. Are there any researches that study the correlation between diet and development of GU? Are there any clinical trials that study how the gut microbiome chemically modify the drugs?
There are some typos in the manuscript. For example, there are 2 consecutive "malignancies" at line 20; "this" and "the" at the end of line 355 and beginning of line 356.
Author Response
Reviewer 3:
Comment 1:
Your manuscript focuses on the treatment of genitourinary cancers. Are there any researches that study the correlation between diet and development of GU? Are there any clinical trials that study how the gut microbiome chemically modify the drugs?
Response 1:
Thank you, additional information regarding diet and development of GU cancers and information regarding gut microbiome and drug modification were added in sections 3 and 5 respectively.
Comment 2:
There are some typos in the manuscript. For example, there are 2 consecutive "malignancies" at line 20; "this" and "the" at the end of line 355 and beginning of line 356.
Response 2:
Thank you we have fixed these typos.
Round 2
Reviewer 2 Report
Comments and Suggestions for Authors
All comments are solved.
No further queries
Comments on the Quality of English LanguageNo comments
Reviewer 3 Report
Comments and Suggestions for Authors
The revised manuscript looks much better. I don't have any questions or concerns.